# Metabolomics of Non-*Saccharomyces* Yeasts in Fermented Beverages

Daniel J. Ellis [1], Edward D. Kerr [1], Gerhard Schenk [1,2] and Benjamin L. Schulz [1,*]

1 School of Chemistry and Molecular Biosciences, The University of Queensland, St Lucia, QLD 4072, Australia; daniel.ellis@uq.edu.au (D.J.E.); edward.kerr@uq.edu.au (E.D.K.); schenk@uq.edu.au (G.S.)
2 Australian Institute of Bioengineering and Nanotechnology, The University of Queensland, St Lucia, QLD 4072, Australia
* Correspondence: b.schulz@uq.edu.au

**Abstract:** Fermented beverages have been consumed for millennia and today support a global industry producing diverse products. *Saccharomyces* yeasts currently dominate the fermented beverage industry, but consumer demands for alternative products with a variety of sensory profiles and actual or perceived health benefits are driving the diversification and use of non-*Saccharomyces* yeasts. The diversity of flavours, aromas, and other sensory characteristics that can be obtained by using non-*Saccharomyces* yeasts in fermentation is, in large part, due to the diverse secondary metabolites they produce compared to conventional *Saccharomyces* yeast. Here, we review the use of metabolomic analyses of non-*Saccharomyces* yeasts to explore their impact on the sensory characteristics of fermented beverages. We highlight several key species currently used in the industry, including *Brettanomyces*, *Torulaspora*, *Lachancea*, and *Saccharomycodes*, and emphasize the future potential for the use of non-*Saccharomyces* yeasts in the production of diverse fermented beverages.

**Keywords:** non-*Saccharomyces* yeasts; fermented beverages; metabolomics; mass spectrometry; sensory characteristics



## 1. Introduction

Fermented beverages have been consumed since the emergence of civilisation. The earliest evidence of purposeful production and consumption of fermented beverages dates back as early as 7000 BC in China [1] and 6000–5800 BC in Georgia [2]. Although fermentation has used diverse microorganisms throughout history, yeasts have been and still are by far the most widely used microorganism in the production of fermented beverages [3,4].

It is likely that the first fermented beverages were made fortuitously due to spontaneous fermentation by microbial contaminants during storage of surplus goods. The benefits of fermentation, including new sensory properties, tastes, aromas, mouth-feel, the analgesic and euphoric effects of ethanol, and protection from further contamination by pathogenic microbes may have triggered systematic efforts to replicate and optimise fermentation [5,6]. These efforts would have likely involved back-slopping, the addition of small amounts of a fermented end-product to the next unfermented batch, beginning the domestication of yeasts and other microbes. Although *Saccharomyces* yeasts are not particularly abundant in the wild [7], their dominance in these controlled fermentations has led to their selection and use for most industrial fermentation processes today, including beer and wine production [5,8,9], either as the sole fermentative microbe or in conjunction with others such as bacteria for kombucha [10,11] or moulds for sake [12,13].

Beer brewing and wine making produce diverse and complex products from a few relatively simple starting ingredients: grapes for wine [14], and barley and hops for beer [15,16]. While these ingredients are key in determining the characteristics of the end product, the yeast strain used has a critical impact on the style, flavour, aroma, and other sensory characteristics of the final product [17]. Modern beer production predominantly uses

domesticated *Saccharomyces cerevisiae* (for ales) or *S. pastorianus* (for lagers) due to their efficient growth on maltose, high ethanol production, attractive flavour profiles [6], and efficient flocculation [18]. In contrast to beer production, modern wine production still largely relies on the autochthonous (indigenous) yeast present on grapes to carry out fermentation, with the addition of *S. cerevisiae* ensuring efficient fermentation and consistent final products [19,20].

Alcoholic beverages such as beer, wine, and cider are the most economically important fermented beverages, with the US beer market alone being valued at US$94 billion in 2020 [21]. However, sales of conventional ale and lager style beers are currently shrinking in most markets. In contrast, there has been recent notable growth in non-alcoholic fermented beverages. For instance, due to growing consumer interest in healthy alternative products to traditional juices and carbonated beverages, the market for products like kombucha has dramatically expanded, with the kombucha market valued at US$1.67 billion in 2019 and expected to grow to US$7.05 billion by 2027 [22]. Consumer demand for alternative products has also driven the craft beer industry over the last decade, with independent breweries contributing US$74 billion and creating more than half a million jobs in 2018 in the USA market alone [23].

Despite the prevalence of *Saccharomyces* yeasts in industrial fermentation [5,8,17], there is increasing interest in non-*Saccharomyces* yeasts, triggered by their rich and diverse reservoir of enzymes and secondary metabolites not typically produced by representative *Saccharomyces* yeasts (Figure 1). The metabolic diversity provided by non-*Saccharomyces* yeasts has promise for meeting consumer demands for new sensorial and beneficial health properties, but without the challenges of mixed microbial ferments. In concert, metabolomics technologies have provided essential data to measure and evaluate the molecular impact and potential of non-*Saccharomyces* yeasts and their metabolites in fermented beverages. This mini-review addresses recent developments in the use of metabolomics to study the use of non-*Saccharomyces* yeasts in the production of fermented beverages.

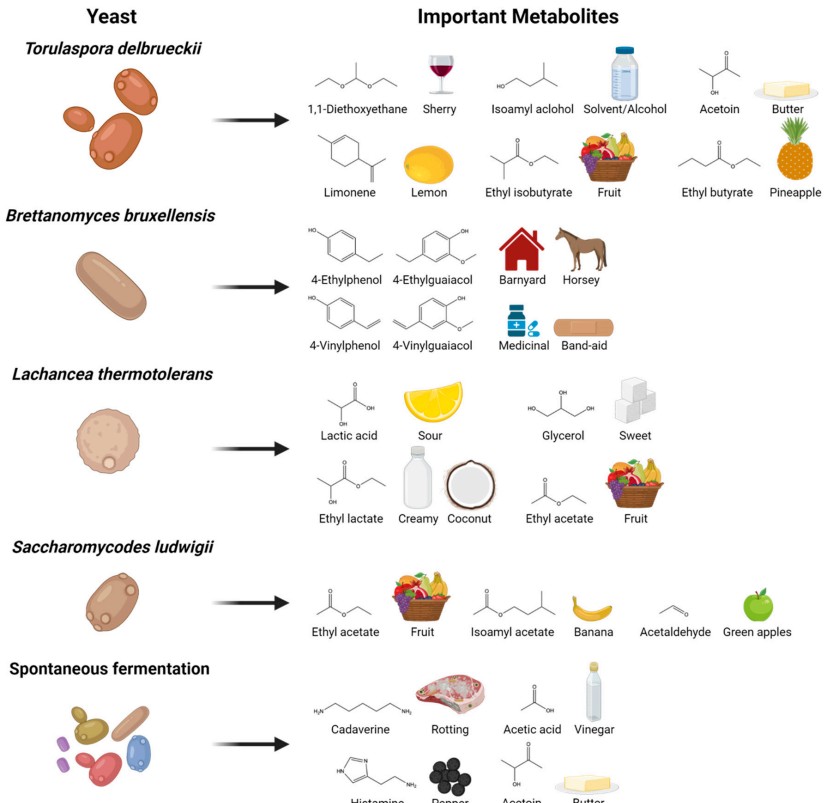

**Figure 1.** Common non-*Saccharomyces* yeasts used in beverage fermentation and their important metabolites.

## 2. Metabolomics Techniques

Metabolomics, the global analysis of low-molecular-weight compounds in complex biological systems, has become a powerful and widely applied approach for studying metabolism in diverse applications including in disease, biomarker identification, drug discovery, and microbial interactions [24]. The key advantage of metabolomics is its ability to measure the global chemical complexity of a system, rather than focussing on individual components. However, metabolomics is uniquely challenging compared to other omics methods such as genomics or proteomics, due to the extreme chemical diversity of the metabolites.

Metabolomics typically uses nuclear magnetic resonance (NMR), liquid chromatography mass spectrometry (LC-MS), or gas chromatography-MS (GC-MS) [24]. Recent advances in instrument performance, including chromatographic separation, resolution, sensitivity, and detection speed, now allow qualitative and quantitative analyses of thousands of metabolites from a sample [25]. However, currently, no single technique can identify or quantify all metabolites due to their extreme chemical diversity and their wide concentration range and because not all metabolites are currently known. Ongoing advances in bioinformatic tools and growing public libraries of metabolite spectra are helping to alleviate some of these problems [25]. While NMR, LC-MS, and GC-MS are all used in metabolomics for diverse applications [24] and specifically for studying yeasts [25], MS-based methods are generally preferred. This is because MS is more sensitive than NMR and can be easily coupled with various chromatography methods to allow separation and quantification of diverse metabolites [24].

Several types of mass spectrometers are popular for metabolomics, each with distinct advantages. Triple quadrupoles are highly sensitive and selective, making them well suited for quantification and targeted metabolomics [24]. However, their low resolution limits their use in untargeted analyses [24,26]. Fourier-transform ion cyclotron resonance (FT-ICR) mass spectrometers provide exceptionally high resolution, but can be difficult and costly to operate, which leads to their limited use in metabolomics [24,26]. Orbitrap mass spectrometers allow very high mass accuracy and resolution and are very useful for untargeted metabolomics [24]. Quadrupole Time-of-Flight (QToF) mass spectrometers provide high mass accuracy and resolution, and are popular for metabolomics because of their robustness and broad utility for quantification and identification in untargeted metabolomics approaches [24].

Detection sensitivity and quantification are further improved when mass spectrometry is coupled with in-line chromatographic separation techniques such as gas-chromatography (GC) or liquid-chromatography (LC) [24,26]. GC is typically used for volatile non-polar metabolites, but sample derivatization can expand the range of metabolites that can be detected by GC-MS to include more polar and less volatile compounds such as fatty acids, amino acids, amines, sterols, and sugars [27]. This allows for the detection of important metabolites relevant to the sensory profile of beverages, such as fatty acids, including lactic acid and acetic acid, which are important in sour beers and wines. LC for metabolomics typically uses either reversed-phase LC (RPLC) or hydrophilic interaction LC (HILIC) [24,27]. RPLC allows separation of more hydrophobic compounds such as phenolic acids, flavonoids, alkaloids, peptides, and certain glycosylated species [28], while HILIC can separate polar compounds such as amino acids, sugars, and nucleotides [28]. In the context of fermented beverages, GC-MS and LC-MS allow complementary analysis of the metabolites that are important contributors to beverage quality (Figure 2 and Table 1).

While metabolomics is relatively mature, its range of applications is still growing. In particular, metabolomics is emerging as a powerful tool for the assessment of quality, flavour, aroma, and sensorial characteristics in the food and beverage industry, especially for fermented beverages with complex metabolite profiles originating from diverse organisms including yeasts.

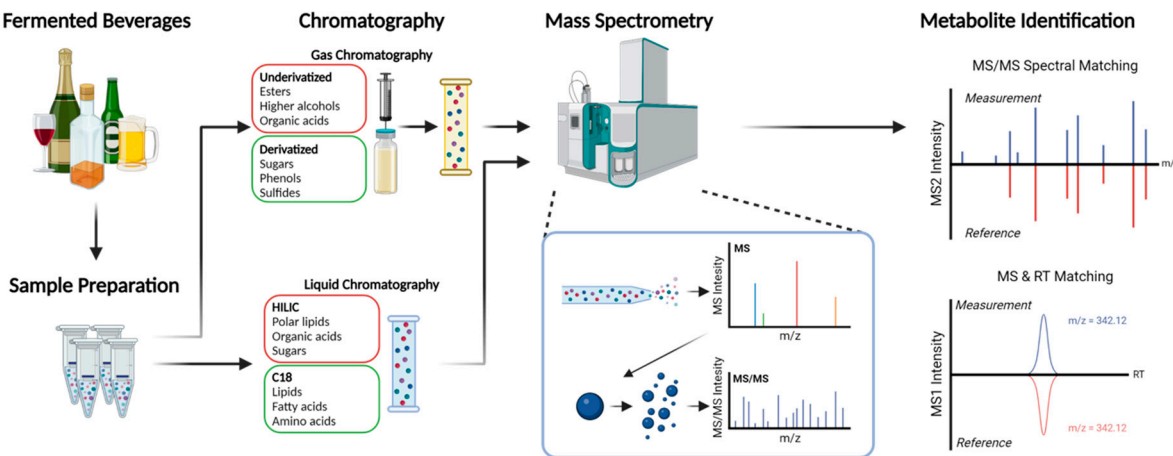

**Figure 2.** Typical workflows for metabolomic analysis of fermented beverages. Key steps include sample preparation, chromatographic separation, mass spectrometry, and data analysis based on database entries or pure standards.

**Table 1.** Analytical methods suitable for analysis of important metabolites produced by non-*Saccharomyces* yeasts.

| Yeast | Metabolite of Interest | Analytical Method Used | Reference |
|---|---|---|---|
| *Torulaspora delbreuckii* | 1,1-Diethoxyethane | SBSE-TD-GCMS | [29] |
| | Limonene | SBSE-TD-GCMS | |
| | Isoamyl alcohol | GC-FID | |
| | Acetoin | GC-FID | |
| | Ethyl isobutyrate | SBSE-TD-GCMS | |
| | Ethyl butyrate | SBSE-TD-GCMS | |
| *Brettanomyces bruxellensis* | 4-Ethylphenol | SPME-GCMS | [30] |
| | 4-Ethylguaiacol | SPME-GCMS | |
| | 4-Vinylphenol | SPME-GCMS | |
| | 4-Vinylguaiacol | SPME-GCMS | |
| *Lachancea thermotolerans* | Lactic acid | HPLC-RI | [31] |
| | Ethyl lactate | HS-SPME-GCMS | |
| | Ethyl acetate | HS-SPME-GCMS | |
| | Glycerol | HPLC-RI | |
| *Saccharomycodes ludwigii* | Ethyl acetate | SPME-GCMS | [32] |
| | Isoamyl acetate | SPME-GCMS | |
| | Acetaldehyde | SPME-GCMS | |
| Spontaneous fermentation | Cadaverine | UPLC-MS/MS | [33] |
| | Acetic acid | HPLC-RI | |
| | Histamine | UPLC-MS/MS | |
| | Acetoin | GC-FID | |

## 3. Non-*Saccharomyces* Yeasts in Beer and Wine

While many non-*Saccharomyces* yeasts have been investigated for their use in the production of fermented beverages, yeast of the genera *Brettanomyces*, *Lachancea*, *Torulaspora*, and *Saccharomycodes* have seen the greatest interest. Spontaneous or mixed ferments are also increasingly of interest in a consumer market looking for a variety of flavours, aromas, and other unusual sensory characteristics. Ethanol and carbon dioxide are the main metabolites produced by yeasts during fermentation, and, while these are responsible for key aspects of the sensory characteristics of fermented beverages, yeasts also produce hundreds of secondary metabolites which impact the sensory profiles of fermented beverages [9]. Secondary metabolites such as esters and higher alcohols, which are regularly associated

with fruity descriptors, often make important contributions to the flavour and aroma of beer and wine [34,35]. Vicinal diketones are another group of important secondary metabolites, such as diacetyl, which imparts unwanted buttery flavours when present in high concentrations [34,35]. The following sections review the use of yeasts from these genera and the metabolites they produce which impact the sensory characteristics of controlled and spontaneous fermentations.

### 3.1. Brettanomyces

*Brettanomyces* yeasts, also known as *Dekkera*, are the most common non-*Saccharomyces* yeasts used by the brewing and wine-making industries. *Brettanomyces* are traditionally considered to be wine spoilage yeasts [36,37], particularly due to their production of 4-ethylguaiacol (4-EG) and 4-ethylphenol (4-EP), with flavours described as medicinal, barnyard, horsy, earthy, spicy, and clove-like [36]. *Brettanomyces* spoilage is still a critical issue in the wine industry today [36,38,39]. Despite its status as a spoilage yeast, the presence of *Brettanomyces* is required in fermentation of certain specialty beers such as Berliner Weisse, lambic, and sour English beers to obtain their characteristic complex "Brett" flavour [30]. In wine, the "Brett" character is associated with negative descriptors such as mousey and metallic [39] but also with positive descriptors such as fruity, tropical, and floral [39]. 4-EP and 4-EG are the major metabolites responsible for "Brett" character, but several volatile ethyl esters including ethyl acetate, ethyl lactate, ethyl caprate, and ethyl caprylate contribute to the fruity and floral descriptors [39]. The difference in perception of the "Brett" character between beer and wine is heavily influenced by the relative concentrations of 4-EG and 4-EP, where 4-EG is higher in beer and 4-EP is higher in wine [39,40].

The use of *Brettanomyces* yeasts for producing beers with distinctive aromas and flavours has recently attracted increased attention, in particular with two species: *B. bruxellensis* and *B. anomalus*. Colomer et al. [40] assessed the population and genetic diversity of 84 *Brettanomyces* yeast species isolated from different fermentation sources across various locations and identified a phenolic-off-flavour-negative *B. anomalus* strain that produced no 4-EP, minimal 4-EG, and no detectable amounts of the intermediates 4-vinylphenol (4-VP) or 4-vinylguaiacol (4-VG). The *B. anomalous* strain is, therefore, a promising strain for the production of beers with characteristic "Brett" flavours but without overwhelming phenolic off-flavours.

In 2017, Crauwels et al. [30] used eight *B. bruxellensis* strains isolated from different fermented beverages, beer, wine, and soft drinks in small scale fermentations of wine (red and white), beer (Duvel and blonde ale), and a soft drink (ginger ale), all supplemented with additional maltooligosaccharides, to test how media composition and strain affected the aromatic profiles of the fermented products [30]. All strains could grow in beer and soft drink media, but only the strains initially isolated from wine were able to grow in the wine media, due to their high sulfite tolerance [30]. Aromatic profiles of the different ferments identified 84 volatile compounds in the test media prior to inoculation and an additional 12 following fermentation. Changes in the volatile compositions of the ferments were both strain- and media-dependent. For example, while volatile phenols and ketones increased in all media, ethyl esters only increased in Duvel beer, while aldehydes particularly increased in white wine [30]. Production of 4-VG, 4-VP, 4-EG, and 4-EP was also both strain- and media-dependent [30]. These compounds were only detected after fermentation in red wine and strong golden pale ale, demonstrating the dependence on media, with only the strains that could grow in red wine media producing these phenolic compounds, highlighting both strain-dependence and the connections between growth specificities and flavour production [30]. Surprisingly, even though these *B. bruxellensis* strains had distinct differences in flavour profiles and growth preferences, sequencing of their DbPAD gene, encoding a phenolic acid decarboxylase, and DbVPR/SOD gene, encoding an enzyme with both vinyl phenol reductase and superoxide dismutase activities, found minimal coding

differences between the strains [30]. This highlights the importance of phenotypic analyses, including metabolomics, for strain comparison and selection.

### 3.2. Lachancea

With the rise of consumer interest in sour beers, yeasts of the genus *Lachancea* have received increased attention because of their ability to produce both ethanol and lactic acid. To produce sour beers, many breweries have turned to using lactic acid bacteria (LAB) in a process called kettle souring. However, this comes with the disadvantage of complicating the cleaning process of brewing equipment to avoid the cross contamination of ferments of other beer styles with LAB. Yeasts of the genus *Lachancea* thus represent a novel bacteria-free method for producing sour beers. *L. thermotolerans* is the most studied species of the genus, with a focus on its ability to acidify beer and wine and additionally impact aroma and flavour [41–43]. *L. fermentati*, while not as thoroughly studied, has also been explored for the production of low-alcohol beers [44,45].

The use of *L. thermotolerans* to acidify wine is of particular interest for wines produced from grapes grown in warmer climates with high sugar content and low organic acid content. Traditionally, the problem of low acidity would be addressed by the addition of tartaric acid [46], but this is costly and can affect the quality of the wine [31]. In 2021, Hranilovic et al. [31] built on their previous work with *L. thermotolerans* [47,48] to study the impact of this yeast on the chemical composition and sensory properties of Merlot wines. The authors carried out fermentations with five *L. thermotolerans* strains in co-inoculated and sequentially inoculated ferments with *S. cerevisiae*. HPLC was used to measure glycerol, lactic acid, malic acid, and acetic acid, while HS-SPME GC-MS was used to assess the presence and concentration of volatile secondary metabolites in the finished wines. Sequential ferments using first *S. cerevisiae* and then *L. thermotolerans* resulted in the lowest ethanol and highest lactic acid concentrations, but co-inoculation of *S. cerevisiae* and *L. thermotolerans* still produced moderate amounts of lactic acid. Strain differences between the lactic acid production of different *L. thermotolerans* strains was also observed. A total of 31 volatile compounds were identified in the wines, which were predominantly yeast-derived metabolites [31]. These metabolites included ethyl esters, acetate ester, higher alcohols, organic acids, and linalool. Ethyl acetate and ethyl lactate, which both contribute to 'fruity' aromas and complexity in wine, were the predominant esters in the wines. The *S. cerevisiae* monoculture wine had the lowest concentration of ethyl acetate, with up to 2.5 times higher concentrations found in sequential fermentations. These higher concentrations did not, however, exceed the threshold at which ethyl acetate becomes undesirable rather than contributing a 'fruity' complexity. These results highlight the effective use of *L. thermotolerans* in sequential fermentations with *S. cerevisiae* over co-inoculation or uninoculated spontaneous fermentation methods to achieve lower ethanol content and better acidity in Merlot wines and wines in general.

### 3.3. Lambic and Mixed-Culture Beer

Spontaneous fermentation is the process by which wort or must is inoculated by native microbes present in the air, rather than by the purposeful addition of yeast. As a result of the secondary metabolites and fermentation end-products produced by these bacteria and yeast, beers produced in this fashion are diverse in character, with flavours and aromas ranging from smooth and tart to complex funk and sour. Lambic beers are the most traditional form of spontaneously fermented beer, brewed around Brussels in the Pajottenland valley of the Senne river. Interest in the production of beer in this style is increasing worldwide, and American craft breweries have implemented similar methods for producing American coolship ales [49]. The production of these styles of beer is like any other, with the exception that fresh boiled wort is allowed to cool overnight in open vessels, which allows it to be inoculated by native yeasts and bacteria, before it is transferred to barrels or casks for fermentation over months or years.

The changes in microbial community composition during the four distinct stages of fermentation of beers of this lambic style are well understood [49–55]. The first stage of lambic fermentation, the *Enterobacteriaceae* stage, occurs three to seven days into fermentation and is characterised by the abundance of *Enterobacter* spp., a suite of other bacteria, yeasts including *Saccharomyces* spp., and a lack of lactic acid bacteria [50,51,54]. This is followed by the main fermentation stage, beginning three to four weeks into fermentation, which is denoted by the abundance of *Saccharomyces* species including *S. cerevisiae*, *S. pastorianus*, and *S. uvarum* [49–51,53,54]. The acidification stage occurs three to four months into fermentation and is characterised by the presence of LAB and acetic-acid-producing yeasts. The shift from *Enterobacteriaceae* to LAB during this stage is due to the increased ethanol levels from the main fermentation stage. The final stage, known as the maturation stage, occurs after ten months and is characterised by a decrease in LAB [50,51,54]. While acetic acid bacteria (AAB) are present throughout the fermentation [50,51,53], the final two stages of lambic fermentation are dominated by the presence of ethanol- and acid-resistant, lactic-acid-producing *Brettanomyces* and *Pedicoccus* species [49].

In a 2018 study, De Roos et al. [33] investigated the production of metabolites during lambic beer production, identifying metabolites from both yeasts and bacteria associated with the different stages of fermentation and maturation. The authors followed two lambic beer production processes from a traditional lambic brewery located in the Senne river valley over a period of 24 months. The fermentations were carried out with the same wort, which had been manually acidified by the addition of lactic acid prior to being transferred to two identical 660 L oak casks. Several methods were used to measure the different metabolites present, including HPLC coupled with either refractive index detection for measuring ethanol and short-chain fatty acid concentrations or ultraviolet detection for lactic acid stereoisomers. Volatile compound concentrations were measured using GC coupled with flame ionisation detection, while ultra-performance LC (UPLC) coupled with tandem mass spectrometry (MS/MS) was used to determine organic acid and biogenic amine concentrations.

Limited growth of the background microbiota was observed during the short *Enterobacteriaceae* stage, with an indicative absence of both short- and branched-chain fatty acids. Most metabolites observed during the main fermentation stage were yeast-associated metabolites including ethanol, methyl-1-butanol, and succinic acid. The production of ethyl acetate and ethyl lactate, the most abundant esters in lambic beer, coincided with the beginning of the main fermentation stage at week 3, with maximum concentrations measured during the final maturation stage. Acetoin levels corresponded with the presence and growth of AAB, with production starting and then increasing from 7 weeks to 6 months before dropping, coinciding with a lower AAB cell count, likely due to consumption of acetoin by yeast cells. Acetic acid followed a similar trend to acetoin, with production beginning from week 7 and increasing along with the increase in AAB. However, acetic acid was not consumed and even increased during the acidification and maturation stages, likely due to the presence of *Brettanomyces* species, *P. damnosus*, and *P. membranifaciens*. Both D- and L-lactic acids were produced in near equal concentrations, reaching a maximum in the acidification and maturation stages. The depletion of malic acid coincided with increases in lactic acid bacteria (LAB), consistent with malolactic fermentation, which had not been previously reported for lambic fermentations.

Biogenic amines are potentially dangerous when present in high concentrations [33,56,57], and are typically produced by enterobacteria in the *Enterobacteriaceae* stage of lambic beer production [33,58]. Using targeted metabolomics, it was found that, in addition to carbohydrates and organic acids, the initial wort contained low levels of the biogenic amines agmatine, putrescine, and cadaverine [33]. Two additional biogenic amines, histamine and tyramine, were produced throughout the fermentation and ageing process but biogenic amine levels remained generally low, likely due to manual wort acidification limiting the growth of enterobacteria. Correlating metabolomics and genomics data suggested that histamine was produced by the LAB species *P. damnosus* present during the acidification stage,

although it is likely that other species also contributed, given the complexity of the microbial community. These studies highlight the power of combining genomics and metabolomics to understand the processes of fermentation in complex microbial communities.

### 3.4. Torulaspora

*Torulaspora delbrueckii*, once considered a contaminant yeast in wines, is now one of the most well-studied non-*Saccharomyces* yeasts [59,60], with well-documented use in the production of beer and wine [59–68]. *T. delbrueckii* strains have been considered for use in the production of low-alcohol beers by Canonico et al. [60], who screened 28 strains and observed generally low ethanol production. There has also been extensive research on the use of *T. delbrueckii* to introduce complexity and desirable aromas and flavours to beer [59] and wine [61]. Michel et al. [59] screened 10 *T. delbrueckii* strains and identified 1 strain that produced beer with a desirable fruity flavour, while 2 other strains showed potential for use in bio-flavouring in the pre-fermentation of beer. These two strains were unable to utilise maltose and showed minimal consumption of amino acids and limited ethanol production, but produced a rich fruity flavour, suggesting that they could be used in low-alcohol beer production or pre-fermentation of wort prior to completing fermentation with *Saccharomyces* yeast. Tataridis et al. [61] observed that wines fermented with *T. delbrueckii* scored higher in aroma, complexity, and fruit descriptors when compared to *S. cerevisiae*-fermented wines, as judged by a panel of experienced oenologists. These features correlated with a higher production of 2-phenyl ethanol, responsible for rose-like aromas [61].

More recently, Ogawa et al. [29] analysed what they termed the "minor volatilome" of *T. delbrueckii* to identify metabolites with potential impacts on aroma profiles. This procedure used fermentation in a high-glucose synthetic medium to mimic the conditions typically seen in wort and must while allowing the detection of metabolites produced by the yeast cells at low levels, which would be obscured by a complex growth medium. Two strains of yeast isolated from the Montilla-Moriles region in Spain were compared, *T. delbrueckii* and *S. cerevisiae*. Metabolites related to sensory properties of fermented beverages such as major volatiles and polyols were identified by GC flame ionisation detection, while volatiles in concentrations less than 10 mg/L were measured using a minor volatilome approach involving stir-bar sorptive extraction thermal desorption gas chromatography mass spectrometry (SBSE-TD-GC-MS). The *T. delbrueckii* fermentations had significantly lower volatile acidity than the *S. cerevisiae* fermentations, a desired trait in winemaking where high volatile acidity is associated with vinegar odour and flavour descriptors. The minor volatilome approach identified 24 minor volatile metabolites, 20 of which were significantly different in abundance between the two yeast fermentations. Correlating the abundance of these metabolites with aroma identified the most important compounds as: ethanol, ethyl propanoate, ethyl butyrate, decanal, ethyl isobutyrate, isoamyl alcohol, ethyl heptanoate, 1,1-diethyoxyethane, nonanal, acetaldehyde, ethyl acetate, acetoin, and limonene. Of these, ethyl butyrate, ethyl isobutyrate, ethyl heptanoate, and nonanal were significantly different in *T. delbrueckii* fermentations compared to *S. cerevisiae* fermentations. These compounds are described as sweet or fruity, suggesting that a product fermented with this strain of *T. delbrueckii* would likely possess these desirable aroma descriptors. The minor volatilome approach also allowed, for the first time in *T. delbrueckii* fermentations, the identification of 1,1-diethoxyethane, 2(5H)-furanone, limonene, ethyl heptanoate, dodecanoic acid, and palmitic acid. However, these compounds were not present in sufficiently high concentrations to influence aroma or flavour, though they may provide other benefits such as antimicrobial properties. Together, this highlights the analytical possibilities of the minor volatilome approach, using defined minimal media to allow detection of minor metabolites, especially when integrated with sensory analyses.

### 3.5. Saccharomycodes ludwigii

The market for low and alcohol-free beers has seen significant growth in recent years [69,70]. Classifications for low-alcohol or alcohol-free beer vary globally with coun-

tries such as Italy and France classifying beer with 1.2% *v/v* ethanol as alcohol-free, while in Denmark and the Netherlands alcohol-free beer must contain <0.1% *v/v* ethanol [32]. In the USA, these classifications are separated, with low-alcohol beer containing <0.5% *v/v* ethanol and alcohol-free beers <0.05% *v/v* ethanol [32].

Most methods to produce alcohol-free beer, such as distillation to remove ethanol after fermentation, are costly and impact the organoleptic properties of the beverage, resulting in beers with little depth of flavour, aroma, or character [32,69–71]. Biological methods of reducing the ethanol content are therefore promising alternatives. One such biological method is the use of non-*Saccharomyces* yeasts with a limited ability to produce ethanol during fermentation. *Saccharomycodes ludwigii* is one such yeast, as it cannot efficiently consume maltose [5,32].

De Francesco et al. [32] screened six strains of *S. ludwigii* and assessed the production of ethanol and volatile compounds during wort fermentation to determine their suitability to produce low-alcohol beer. The metabolite profiles of the resulting beers were analysed using SPME-GC-MS, with aldehydes and vicinal diketones derivatized, enabling detection. All strains produced low ethanol ferments ranging from 0.51–1.36% *v/v*, with a total sugar consumption of less than 2 kg/hL [32]. The volatile profiles of the final beers showed substantial strain-specific variability, with orders-of-magnitude differences between strains in the abundance of the total amount of esters, higher alcohols, aldehydes, and vicinal diketones. *S. ludwigii* strain DBVPG 3010 was identified as particularly interesting for the production of low-alcohol beer due to its limited ethanol production and ability to produce appreciable levels of both esters and higher alcohols in the absence of off-flavour metabolites above their perception threshold.

### 3.6. Other Non-Saccharomyces Yeasts

In addition to the non-*Saccharomyces* yeasts described in detail above, many other non-*Saccharomyces* yeasts have been explored for their potential impacts on the sensory properties of fermented beverages. Most non-*Saccharomyces* yeasts that have been used for the production of fermented beverages belong to the following genera: *Brettanomyces*, *Candida*, *Debaryomyces*, *Hanseniaspora*, *Kazachstania*, *Kluyveromyces*, *Lachancea*, *Metschnikowia*, *Meyerozyma*, *Pichia*, *Rhodotorula*, *Starmerella*, *Saccharomycodes*, *Saccharomycopsis*, *Torulaspora*, *Trichosporon*, *Wickeramomyces*, *Williopsis*, *Yarrowia*, *Zygoascus*, and *Zygosaccharomyces* [72]. As an example, *Cyberlindnera* (formally *Williopsis*) *saturnus* produces isoamyl acetate, which imparts fruity flavours reminiscent of apple, pear, and banana [73]. In contrast, while *P. kluyverii* also produces high levels of isoamyl acetate, this is accompanied by high levels of ethyl acetate, which gives an undesirable solvent-like aroma [74]. The production of low- or non-alcoholic beers is another area in which the use of non-*Saccharomyces* yeasts is of great interest. Osburn et al. [42] identified five species of yeasts from four genera which could produce lactic acid alongside ethanol, three of which were not *Lachancea* yeasts (*Hanseniaspora vineae*, *Schizosaccharomyces japonicus*, and *Wickerhamomyces anomalus*). Rodríguez Madrera et al. [75] evaluated eight non-*Saccharomyces* yeast strains (*S. ludwigii*, *Metchnikowia pulcherrima*, *H. uvarum*, *H. osmophila*, and *B. bruxellensis*) isolated from cider for their use in brewing. Yeasts isolated from kombucha (*H. valbyensis*, *H. vineae*, *T. delbrueckii*, *Z. bailii*, and *Z. kombuchaensis*) have been investigated for use in producing non-alcoholic beer [76]. Canonico et al. [62] examined the volatile profiles of wines produced using *M. pulcherrima*, *T. delbrueckii*, and *Z. bailii* as a means of reducing ethanol concentrations. Binati et al. [20] studied the contribution of three non-*Saccharomyces* yeasts (*L. thermotolerans*, *Metschnikowia* spp., and *Starmerella bacillaris*) to volatile and sensory profile diversity in sequential inoculation of wine with *S. cerevisiae*, finding that *S. bacillaris* increased glycerol while lowering acetaldehyde and total sulphur content, while *Metschnikowia* spp. was linked with greater higher alcohol and ester formation [20]. Together, these observations highlight the potential in investigating the diversity of non-*Saccharomyces* yeasts and their complex metabolomes for use in fermented beverage production.

## 4. Current Issues and Emerging Directions

Non-*Saccharomyces* yeasts present exciting opportunities in the fermented beverage industry for creating varied and diverse products to meet emerging consumer demands. However, despite the growing body of research supporting the use of non-*Saccharomyces* yeasts to achieve desired characteristics in fermented beverages, their use is not yet commercially widespread. This slow uptake may be due to a lack of extensive characterization of the impacts of different species and strains of non-*Saccharomyces* yeasts on the organoleptic properties of fermented beverages. Major challenges constraining the characterization of non-*Saccharomyces* yeasts and their use in fermented beverage production are issues of data availability and curation within metabolite databases and a lack of a thorough understanding of how metabolite profiles affect beverage sensory profile.

Perhaps the greatest hurdle for the widespread molecular characterization of the metabolomes of non-*Saccharomyces* yeasts is a lack of accessible, well-curated metabolite databases. Identification of metabolites using untargeted MS approaches typically relies on matching experimental data with known database entries based on molecular mass, characteristic MS/MS fragment ion abundances, and associated data such as retention time (Figure 2). While several high-quality curated MS databases are available, including NIST, METLIN, mzCloud, MassBank, and HMDB, these databases only contain MS/MS ion fragment spectra for a small fraction of the more than 100 million compounds listed in databases such as PubChem [24]. Substantial bias also exists in the types of compounds present in databases, with poor coverage of rare or low-abundance metabolites. There is also a bias towards spectra acquired by GC-MS, because of its earlier development. This is particularly problematic because LC-MS has emerged as a popular preferred technique for metabolomics, with several different MS instrument designs and fragmentation techniques. It is therefore vital that the community continues submission and curation of well-annotated data generated with a variety of instruments and methods, including the submission of newly discovered compounds. Unfortunately, community-generated datasets have their own potential problems, chief among which for metabolomics data are inconsistent naming conventions. To combat these issues, the Metabolomics Standards Initiative recommended in 2007 that a central repository for metabolite data with strict submission guidelines be created [24,77]. In the time since, recommendations have been made for metabolite data to be submitted with associated InChI and PubChem IDs [78,79] as these are unique and machine-readable naming systems [24]. Unfortunately, these suggestions are yet to be widely adopted.

Correlations or mechanistic connections between sensory analyses and metabolomic profiles of non-*Saccharomyces*-produced beverages are very limited. Studies including both types of analyses are also heavily biased towards wine. Sensory analyses are mostly conducted in parallel with metabolomic analyses, with only limited correlations between the datasets. This is understandable, given the complexity of the metabolomes of fermented beverages and the complex interactions between metabolites that can impact sensory attributes. Direct links between metabolomic and sensory data could be made by identifying correlations in large, diverse, and well-curated datasets, or by experimental manipulation of the metabolomic profiles of beverages coupled with comparative sensory analyses. Neither of these approaches are simple and they are also compounded by the problem of 'matrix' effects, in which a metabolite may have a certain organoleptic attribute in isolation which is, however, altered when present in a complex mixture such as a fermented beverage.

Finally, it is important to consider the most efficient source of new diversity in non-*Saccharomyces* yeasts with potential use in fermented beverage production. Bioprospecting for yeast diversity is possible in natural environments such as plants and associated material and in industrial fermentative environments such as the production of beer, wine, and bread dough [80,81]. However, there are also clear opportunities for the identification of additional diversity in yeasts from traditional fermented beverages such as fermented goats' milk [82], motoho [83], pito [84,85], Daqu [86], khadi [87], and chicha [88], to name just a few. These traditional fermented beverages are likely an excellent source of yeast

with diverse impacts on sensorial properties as they represent a diverse range of flavour and aroma characteristics and are also likely to contain yeasts well suited to industrial fermentative environments. Detailed characterization of the genomic and metabolomic diversity of the yeasts present during production of these beverages would likely provide helpful insights in understanding their novel organoleptic impacts.

While the field is still young and has hurdles to overcome, the metabolomic characterization of non-*Saccharomyces* yeasts has already proven to be a valuable tool for finding alternative ways to meet consumer demands for novel flavours and sensory attributes in fermented beverages. Integrating yeast isolation and traditional microbiology with quantitative sensory evaluation and modern metabolomics will provide exciting opportunities to help develop non-*Saccharomyces* yeast as a viable and growing option for the fermented beverage industry.

**Author Contributions:** Conceptualization, D.J.E., E.D.K., G.S. and B.L.S.; writing—original draft preparation, D.J.E., E.D.K., G.S. and B.L.S.; writing—review and editing, D.J.E., E.D.K., G.S. and B.L.S.; visualization, D.J.E., E.D.K., G.S. and B.L.S.; supervision, E.D.K., G.S. and B.L.S. All authors have read and agreed to the published version of the manuscript.

**Funding:** This research received no external funding.

**Institutional Review Board Statement:** Not applicable.

**Informed Consent Statement:** Not applicable.

**Conflicts of Interest:** The authors declare no conflict of interest.

## Abbreviations

Liquid chromatography mass spectrometry (LC-MS), gas chromatography mass spectrometry (GC-MS), nuclear magnetic resonance (NMR), Fourier-transform ion cyclotron resonance (FT-ICR), Quadrupole Time-of-Flight (QToF), reversed phase LC (RPLC), hydrophilic interaction LC (HILIC), 4-ethylguaiacol (4-EG), 4-ethylphenol (4-EP), 4-vinylguaiacol (4-VG), 4-vinylphenol (4-VP), lactic acid bacteria (LAB), acetic acid bacteria (AAB), high-performance LC (HPLC), ultra-performance LC (UPLC), tandem mass spectrometry (MS/MS), stir-bar sorptive extraction thermal desorption GC-MS (SBSE-TD-GC-MS), solid-phase microextraction (SPME).

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
