# Peer review of "Metabolomics of Non-Saccharomyces Yeasts in Fermented Beverages"

_beverages, doi:10.3390/beverages8030041_

Round 1

Author Response

We thank the reviewer for their comments and suggestions. Detailed responses to each comment are included below.

Comment:

This review is interesting because it highlights the use of metabolomic analyses on the characterization of aromatic substances produced by non-Saccharomyces yeasts currently used in the beer and wine industry.

However, this review lacks a summary table that relates the major metabolites produced by non-Saccharomyces to chromatographic or other techniques using mass spectrometry.

The title of the article is centered on metabolomics and the authors put forward some technical approaches for targeted and non-targeted metabolomic studies, and apart from the two summary figures, no numerical support is accessible as range of concentrations, limit of detection, etc.

For all these reasons, the manuscript may be accepted if the authors add summary tables to the manuscript and respond to the reviewer's comments.

Response:

We thank the reviewer for this helpful comment. We have revised the manuscript by including an additional Table to highlight the connections between the metabolites produced by various yeasts and the analytical techniques used for their measurement.

Table 1. Analytical methods suitable for analysis of important metabolites produced by non-Saccharomyces yeasts.

Yeast

Metabolite of Interest 

Analytical Method Used

Reference

Torulaspora delbreuckii

1,1-Diethoxyethane

SBSE-TD-GCMS

[29]

Limonene

SBSE-TD-GCMS

Isoamyl alcohol

GC-FID

Acetoin

GC-FID

Ethyl isobutyrate

SBSE-TD-GCMS

Ethyl butyrate

SBSE-TD-GCMS

Brettanomyces bruxellensis

4-Ethylphenol

SPME-GCMS 

[30]

4-Ethylguaiacol

SPME-GCMS 

4-Vinylphenol

SPME-GCMS 

4-Vinylguaiacol

SPME-GCMS 

Lachancea thermotolerans

Lactic acid

HPLC-RI 

[31]

Ethyl lactate

HS-SPME-GCMS

Ethyl acetate

HS-SPME-GCMS

Glycerol

HPLC-RI

Saccharomycodes ludwigii

Ethyl acetate

SPME-GCMS

[32]

Isoamyl acetate

SPME-GCMS

Acetaldehyde

SPME-GCMS

Spontaneous fermentation

Cadaverine

UPLC-MS/MS

[33]

Acetic acid

HPLC-RI

Histamine

UPLC-MS/MS

Acetoin

GC-FID

Minor comments.

Abstract.

4-vinylguaiacol

Lachancea

Response: We have corrected these typographical errors.

Comment:

“HPLC was used to measure glycerol, lactic acid, malic acid, and acetic acid”. Lachancea forms L-lactic acid from sugars, with HPLC analysis can you separate the two isomers of lactic acid, D and L?

You quote in the manuscript "Sequential ferments using first S. cerevisiae and then L. thermotolerans resulted in the lowest ethanol and highest concentrations of lactic acid", I think it is necessary to specify the isomerism of lactic acid for Lachancea because lactic acid bacteria also produce L-lactic acid from L-malic acid.

Response: We thank the reviewer for this comment.  We agree that distinguishing between L- and D- forms is analytically critical.  However, as this was not highlighted in the relevant study we have not further emphasised this at this point in the review.

Reviewer 2 Report

The review is interesting. I suggest some minor changes. It should be interesting to prepare a paragraph concerning the main metabolites produced by yeasts and their impact on the analysed beverages. Moreover, the authors should specify why they focus their attention of those species. Finally, the effect of non-Saccharomyces yeasts in mixed fermentation should be better underlined.

Author Response

We thank the reviewer for their suggestions. To address these suggestions, we have revised the manuscript and included additional text in Section 3:

"While many non-Saccharomyces yeasts have been investigated for their use in the production of fermented beverages, yeast of the genera Brettanomyces, Lachancea, Torulaspora, and Saccharomycodes have seen the greatest interest. Spontaneous or mixed ferments are also increasingly of interest in a consumer market looking for a variety of flavours, aromas, and other unusual sensory characteristics. Ethanol and carbon dioxide are the main metabolites produced by yeasts during fermentation, and while these are responsible for key aspects of the sensory characteristics of fermented beverages, yeasts also produce hundreds of secondary metabolites which impact the sensory profiles of fermented beverages [9]. Secondary metabolites such as esters and higher alcohols, which are regularly associated with fruity descriptors, often make important contributions to the flavour and aroma of beer and wine [34,35]. Vicinal diketones are another group of important secondary metabolites, such as diacetyl which imparts unwanted buttery flavours when present in high concentrations [34,35]. The following sections review the use of yeasts from these genera and the metabolites they produce which impact the sensory characteristics of controlled and spontaneous fermentations. "